# The Nonlinear Causal Relationship Between Environmental Regulation and Technological Innovation—Evidence Based on the Generalized Propensity Score Matching Method

**Guan-Yu Zhang, Rong Guan and Hui-Juan Wang \***

School of Statistics and Mathematics, Central University of Finance and Economics, Beijing 100081, China; guanyu-zhang-cufe@outlook.com (G.-Y.Z.); rongguan77@gmail.com (R.G.)

**\*** Correspondence: huijuan-wang-cufe@outlook.com

**Abstract:** The existing research on testing Porter's hypothesis has not considered the selective bias in the sample when establishing a model. However, the selective bias is likely to cause instability of estimation results and reduce the reference value of conclusions. This article, based on individual enterprises in the China Industrial Enterprise Database, aims to verify the selective bias existing in previous research. Then, using the generalized propensity score matching method, a frontier method in the field of causal inference, we re-examined the causal relationship between environmental regulation and two types of technological innovation, weakened endogenous and reverse causal effects, and obtained a more complete and accurate dynamic impact of environmental regulation on the level of technological innovation for enterprises. The main conclusions of this paper are as follows: (1) The influence of environmental regulation on the level of process innovation has two dimensions: time and intensity, and the causal relationship between these dimensions changes from an N shape to an inversed-U shape over time. (2) The influence of environmental regulation on product innovation levels only includes the intensity dimension, and the two produce a U shape. (3) Process innovation and product innovation, to a certain extent, are reflected in the intriguing situation that they cannot gain and lose at the same time. (4) Light industries have a lower tolerance of environmental regulation than heavy industries, and they are more likely to be stimulated by environmental regulation. The conclusions of this paper can provide valuable advice to governments in relation to the formulation of environmental policies and laws.

**Keywords:** environmental regulation; technological innovation; nonlinear causal relationship; selective bias; generalized propensity score matching

## 1. Introduction

General Secretary Xi Jinping pointed out at the 19th National Congress of the Communist Party of China that the Chinese economy has shifted from a high-speed development stage to a high-quality development stage, with the ultimate goal of achieving sustainable economic development. However, with the background of limited input of production factors and serious environmental pollution, extensive economic growth is difficult to sustain. Therefore, the Chinese central government has repeatedly emphasized that it is necessary to transform the mode of economic growth from extensive growth relying on material resource inputs to intensive growth supported by total factor productivity. Although the general direction of transforming the mode of economic growth is clear, there is no conclusion on the specific sustainable development path. Local governments have not achieved good results after changing their modes of economic growth. What is the specific path to achieve sustainable

economic development? What are the levels of "input", "emissions", and "regulation"? These are unresolved questions. Therefore, this article focuses on a specific sustainable development path—the role of environmental regulation in boosting technological innovation. Environmental regulation is the main policy tool to promote the sustainable development of the economy, resources, and society, whose goal is to internalize the social costs caused by corporate emissions, promote corporate energy conservation, reduce emissions, and accelerate technological innovation while achieving environmental protection. Technological innovation is the primary driving force and source of sustainable economic growth. Clarifying the relationship between environmental regulation and technological innovation can help shape a sustainable development path.

Economists of the twentieth century generally held two opposing views on whether environmental regulation can promote technological innovation. Traditional economists believed that environmental regulation would inevitably increase the private cost of enterprises and reduce their market competitiveness. However, Porter proposed the opposite view in the famous Porter hypothesis. According to Porter's hypothesis, appropriate environmental regulation can stimulate regulated enterprises to innovate production technology and production processes, partially or even completely offset compliance costs, and enhance their industrial competitiveness [1]. Porter's hypothesis describes the sustainable state that the country wants to achieve, which is to promote economic development to the maximum extent under the premise of ensuring a high quality of natural resources and providing high quality services. In recent years, several empirical studies have tested the Porter's hypothesis and discussed the impact of environmental regulation on technological innovation or competitiveness, and they can be divided into two distinctly different major categories. The first category is to examine the linear relationship between environmental regulation and the level of technological innovation. Lanoie et al. analyzed more than 4000 companies in seven OECD countries and found that environmental regulation can stimulate technological innovation, and incentivized environmental regulation promotes technological innovation more than command-based environmental regulation [2]. But evidence from the United Kingdom's manufacturing data shows that environmental regulation had no significant impact on total R&D spending [3]. Rennings and Rammer found that environmental regulations in the field of sustainable development stimulated the level of product innovation in an enterprise, whereas environmental regulation in the field of water management did not stimulate it [4]. In addition, many scholars have recently discovered that environmental sustainability orientation is positively affecting corporate competitiveness from the corporate level [5,6]. The second category is to test the nonlinear relationship between environmental regulation and technological innovation. Li and Tao obtained the U-shaped relationship between environmental regulation intensity and technological progress by setting the quadratic term of the environmental regulation's intensity [7]. There are regional differences in the role of environmental regulation in technological innovation, and there is a U-shaped relationship between environmental regulation and the advancement of industrial production technology in eastern and central China, although a statistically significant U-shaped relationship remains to be formed in the west [8]. Johnstone et al. used data from the power industry of 20 countries from 1990 to 2009 to find the inverted U-shaped relationship between the intensity of environmental regulation and technological innovation [9], which means that although stricter emission standards encourage companies to seek efficiency, if these standards are implemented in a differentiated manner, or if regulation intensity is too high, the incentives for environmental regulation to stimulate technological innovation will diminish. An inverted U-shaped relationship between environmental regulation and technological progress was also found based on panel data at the Chinese industrial level [10].

By combing through literary conclusions, we can find that regardless of whether the linear relationship or the non-linear relationship between environmental regulation and technological innovation level is tested, the conclusions drawn by scholars are inconsistent or even contradictory. In the past 20 years, although there have been several studies on Porter's hypothesis, contradictory evidence always existed [11]. However, determining the relationship between environmental regulation

and technological innovation is particularly important for the government to develop environmental policies and laws and explore the path of sustainable development. Specifically, the linear relationship between the two can explain whether the sustainable state of ensuring environmental quality and economic development is possible, that is, whether it is possible to achieve a win-win situation to improve the environment and develop the economy. If the two exhibit a U-shaped nonlinear relationship, the government will be prompted to revise the environmental regulations as soon as possible so that the intensity of environmental regulation breaks through the inflection point of the U-shaped curve and ensure the significance of the U-shaped curve's rising phase. If the two exhibit an inverted U-shaped nonlinear relationship, it indicates that there is an optimal environmental regulation intensity, and the promotional effect of environmental regulation is limited.

This paper argues that the different conclusions of many pieces of literature may be caused by selective bias. Selective bias refers to the selection of the research object that is not random but is based on a certain standard, which will causing the explanatory variables in the regression model to be related to the random disturbance term, thereby triggering more endogenous problems and reverse causal problems in the model [12]. Specifically, the state's implementation of environmental regulations on enterprises is not a random experiment, and enterprises have "self-selected" environmental regulation intensity to a certain extent. In the case of incentivized environmental regulation, the most commonly used environmental regulation incentive is to collect fees for discharging pollution from the enterprise. The enterprise will select the most suitable discharge value according to its own capital status and production status. In this case, this means it will select a specific sewage discharge amount. Suppose a company chooses to pay more sewage charges, which means that the company may have better capital turnover capacity and at a larger scale, but R&D expenditure is closely related to the two. Therefore, in the regression equation, when the intensity of environmental regulation is used to explain technological innovation, the coefficient of the intensity of environmental regulation can be expected to be overestimated. This is because there is a significant correlation between environmental regulation and technological innovation rather than causal relationship, but an ordinary estimation method cannot identify the two-way causal relationship between the two variables. In this case, it is highly difficult to obtain stable results. In the panel regression model, the existence of selective bias often makes the estimation results of regression models with various effects different. Liu and Chen demonstrated that selective bias would cause a correlation of the variables in the time dimension and the cross-sectional dimension to be heterogeneous, while different settings would lead to extensive changes in the estimation results [13]. Although a few scholars introduced instrumental variables into the model to solve the above problems when testing the Porter's hypothesis, traditional instrument variables often fail to accurately estimate parameters in the presence of selective biases [14]. This paper expects to solve the selective bias in the previous research by using the generalized propensity score matching (GPS) method, weakening the endogenous and reverse causal effects, and obtaining a more accurate and stable nonlinear causal relationship between the environmental regulation and technological innovation level, which can provide a feasible path for a country's sustainable development.

The possible innovations in this paper are as follows: First, this article uses causal inference to solve endogenous and reverse causal problems caused by selective bias, namely generalized propensity score matching. This method is an extension of propensity score matching (PSM). Although PSM can resolve selective bias, it cannot describe the dynamic changes of the level of technological innovation of enterprises under different environmental regulations. It does not apply to continuous treatment variables and cannot be used to check the threshold effect. Therefore, this paper selects the GPS method to describe the nonlinear relationship between environmental regulation and technological innovation. This method not only overcomes the selective bias of the sample, but also applies to the continuous treatment variable. Second, this paper explores the sustainable path from the perspective of individual enterprises. Due to being limited by the panel method, the existing research on testing Porter's hypothesis mostly remains on the macro level, while this paper uses individual enterprise data to complete empirical analysis. Third, to prove that the GPS method can effectively solve the selective

bias that exists in the previous research, this paper performs the balance test after implementation of the GPS method.

The structure of this paper is as follows. Section 2 fully describes the GPS method. Section 3 describes the data and variable measures used. Section 4 verifies the existence of selective bias, obtains the nonlinear causal relationship between environmental regulation and two kinds of technological innovation, measures the inflection point values and the number of inflection points, and tests the establishment conditions of Porter's hypothesis. Section 5 presents conclusions and relevant policy recommendations for national sustainable development.

## 2. Methods

The GPS method was first proposed by Hirano and Imbens and is an extension of the PSM method [15]. The PSM method divides the model variables into three categories: treatment variables, matching variables, and result variables, wherein the treatment variables are discrete variables, corresponding to the treatment group and the control group. According to the propensity score, an individual in the control group that is completely similar to the main features of an individual in the treatment group is found, meaning the two groups of samples that are matched differ only in the treatment variables. Furthermore, the control group can approximate the counterfactuals of the treatment group in the hilt. The PSM method is only applicable to treatment variables that are discrete, but there are many cases in which the variables are continuous. Hirano and Imbens extended the PSM method to continuous treatment variables to form a GPS method, and they proposed a method for estimating causal effects in such problems. Irrespective of whether PSM or the GPS method were used, the main idea is to use the propensity score to eliminate selective bias and mixed bias, and more accurately estimate the causal effect of the treatment variables in the result variables. Compared with the ordinary regression model, the GPS method greatly reduces the endogeneity of the model and reduces the standard deviation, making the estimation results more accurate and stable. When using the GPS method to estimate causal effects, the model and variables required to meet the two key assumptions involving conditional independence and balance, is explained in more detail in the following text.

For each individual $i$ in a random sample, suppose there is a set of potential outcomes $Y_i(t)$, the individual's dose response function, wherein $t(t \in D)$ is the treatment variable. For simplicity, the subscript $i$ will be omitted below. The main implication of the conditional independence hypothesis is that, after controlling the difference in covariate $X$, the intensity of treatment accepted by the individual is independent of the underlying outcome. The mathematical expression of conditional independence is (1).

$$Y(t) \perp T \big| X \; \forall \; t \in D \tag{1}$$

Similar to the PSM method, the GPS method uses the conditional density function to estimate the propensity score to control the difference of the covariate $X$, thereby avoiding generating the "high-dimensional curse" when matching multiple covariates. We defined the conditional density function of the treatment variable as:

$$r(t, x) = f_{T|X}(t|x). \tag{2}$$

We defined the generalized propensity score as follows:

$$R = r(t, x). \tag{3}$$

The main meaning of balance is that, after using the generalized propensity score matching method, it is necessary to test that if a group of individuals has the same generalized propensity score, the probability that $T = t$ occurs is independent of the multivariate covariate $X$, that is (4):

$$X \perp 1\{T = t\} \big| r(t, X). \tag{4}$$

Equation (5) combines conditional independence and generalized propensity scores to better illustrate the GPS method. This means that as long as the generalized propensity score is controlled, it not only controls the covariate, but also eliminates the selective bias of the individual in the sample.

$$f_T(t|r(t,x), Y(t)) = f_T(t|r(t,x)) \tag{5}$$

The description above is the theoretical basis of the GPS method. Hirano and Imbens proposed three steps to estimate the causal effect function using the GPS method.

In the first step, the covariate is used to estimate the conditional distribution of the treatment variable $T$ (Continuous variable), while confirming the presence of selective bias. Because the distribution of the treatment variable, i.e., the intensity of the environmental regulation, is right-biased, it is suitable for estimating the conditional distribution of the environmental regulation intensity using the cumulative distribution function of the logistic distribution, namely the Fractional Logit model.

$$E(T_i|X_i) = F(X_i\beta) = \frac{\exp(X_i\beta)}{1 + \exp(X_i\beta)} \tag{6}$$

In (6), $\beta$ is the coefficient vector and $X_i$ is the covariate vector. The maximum likelihood estimation method was used to estimate $\beta$. The probability density of the $i$-th observed individual, that is, the generalized propensity score is:

$$gps_i = \left[F\left(X_i\hat{\beta}\right)\right]^{T_i}\left[1 - F\left(X_i\hat{\beta}\right)\right]^{1-T_i}. \tag{7}$$

In the second step, the conditional expectation value of the result variable $Y_i$ was estimated using the generalized propensity score estimation value obtained in the previous step, and the estimation method was the least square method. The setting of high-order items and interaction items depends on the significance of the model. Based on the results of the parameter estimation, the expected values of the result variables at different levels of treating were obtained.

$$E[Y_i|T_i, gps_i] = a_0 + a_1 T_i + a_2 T_i^2 + a_3 gps_i + a_4 gps_i^2 + a_5 gps_i \times T_i \tag{8}$$

In the third step, the value interval of the treatment variable $T$ was divided into a plurality of subintervals, and $N$ is the number of samples in each subinterval. The expected value of the result variable was estimated in each subinterval using Equation (9).

$$E\left[Y(\hat{m})\right] = \frac{1}{N}\sum_{i=1}^{N}(\hat{a_0} + \hat{a_1}t + \hat{a_2}t^2 + \hat{a_3}gps_i(t, X_i) + \hat{a_4}gps_i(t, X_i)^2 + \hat{a_5}gps_i(t, X_i) \times t \tag{9}$$

Finally, the causal effects of each sub-interval were connected by lines, and the causal relationship between the intensity of environmental regulation and the level of technological innovation of the whole interval was obtained.

## 3. Materials

### 3.1. Variable Description

This article involves three types of variables: treatment variables, matching variables, and result variables. The specific definitions and measures are as follows.

### 3.1.1. Treatment Variable

The treatment variable is the intensity of environmental regulation. The methods for measuring the intensity of environmental regulation in domestic and foreign literature are less uniform, and mainly include the following: major pollutant emissions, wastewater discharge compliance rates,

pollution control costs, the number of environmental laws, per capita GDP, and comprehensive indicators. This paper selects the ratio of sewage charges to the annual output value of industrial enterprises to measure the intensity of environmental regulation. This is mainly because China is now promoting market-oriented options to reduce pollution and emissions. The environmental regulation that this paper hopes to discuss is incentive-type environmental regulation rather than command-based environmental regulation. In addition, referring to the measurement method of Wen and Han [16], according to the characteristics of the Fractional Logit model, the treatment variable was defined as the ratio of the above ratio of the *i*-th enterprise to the maximum value of the ratio in the sample.

### 3.1.2. Result Variables

The result variable is the level of technological innovation that the enterprise can claim credit for. To comprehensively examine the driving role of environmental regulation in technological innovation, Rennings and Rammer discuss the two dimensions of enterprise technological innovation, process innovation and product innovation [4]. This paper refers to this measurement method, used the ratio of R&D expenditure to total sales value to measure the level of process innovation, and measured the level of product innovation by using the ratio of the output value of new products to the total output value.

### 3.1.3. Matching Variables

In the GPS method, selecting the appropriate matching variable can effectively correct the selective bias. A better strategy for selecting matching variables is to select variables that affect both the treatment and result variables. Based on this, this paper refers to Zhou et al. using the PSM method to explore the impact of environmental constraints on R&D expenditures of foreign direct investment enterprises [17], combined with China's method of collecting sewage charges, and after considering the availability of data, selected matching variables. The specific definition and measurements of matching variables are presented in Table 1. To make the data more stable, the value of some variables is logarithmically determined.

In summary, the variables used in this article and the measurement methods are summarized in Table 1.

**Table 1.** Variable description.

| Variable Types | Notation | Variable | Measure |
|---|---|---|---|
| Treatment variable | *T* | The intensity of environmental regulation | Normalized ratio of sewage charges to the annual gross output value of the industrial enterprise |
| Result variables | Process innovation | Level of process innovation | The ratio of R&D expenditures to total sales value |
|  | Product innovation | Level of product innovation | The ratio of the output value of new products to the total value of sales |
| Matching variables | Productivity | Labor productivity | The logarithm of the ratio of the total output value to the total number of employees |
|  | Intensity | Capital intensity | The logarithm of the ratio of the average annual net fixed assets balance to the total number of employees |
|  | Size | The enterprise scale | The logarithm of total sales value |
|  | Age | Enterprise age | The difference between the observation year and the establishment year |
|  | Profit | Profit margins | Ratio of operating profit to total sales value |
|  | Constraint | Financing constraints | The ratio of interest expense to fixed assets |
|  | Cratio | Current ratio | The ratio of interest expense to fixed assets |
|  | Sratio | Debt-equity ratio | The ratio of owner's equity to total debt |

### 3.2. Data

The data used for this paper are from the China Industrial Enterprise Database. The database is a census data of large-scale industrial enterprises at the micro level, including many listed industrial enterprises and non-listed industrial enterprises. It is the most complete and objective data set for non-listed enterprises. In 2015, there were 14,920 enterprises in the list of national key monitoring and discharging enterprises, including 13,555 non-listed companies, accounting for 90.9% of total national enterprises. Therefore, the database is highly suitable for the research questions in this paper. The database provides data for the period between 1998 and 2013. Because the database only disclosed the indicator of sewage charges in 2004 and considering the lag of technological innovation [18], the data period of this paper is from 2004 to 2007. The sewage charges of some industrial enterprises in the database are 0. Because these enterprises are not polluting industrial enterprises or the data are missing, these individuals are not included in this paper. The samples obtained through the complicated cleaning process are 1932 industrial enterprises, and the descriptive statistics of the variables are presented in Table 2.

**Table 2.** Descriptive statistics of variables.

| Variable | Minimum | Mean | Maximum | Standard Deviation |
|---|---|---|---|---|
| Productivity | 2.179 | 5.516 | 8.968 | 0.907 |
| Intensity | 0.248 | 4.399 | 8.123 | 1.016 |
| Size | 6.387 | 11.953 | 17.650 | 1.601 |
| Age | 0.000 | 21.140 | 404.000 | 22.145 |
| Profit | −1.880 | 0.054 | 4.704 | 0.171 |
| Constraint | −0.172 | 0.046 | 1.845 | 0.089 |
| Cratio | 0.079 | 1.729 | 166.924 | 5.707 |
| Sratio | −0.662 | 1.507 | 403.939 | 9.624 |
| T | 0.000 | 0.115 | 1.000 | 0.163 |
| Process innovation (2005) | 0.000 | 0.018 | 1.725 | 0.048 |
| Process innovation (2006) | −0.097 | 0.020 | 1.056 | 0.040 |
| Process innovation (2007) | −0.065 | 0.020 | 0.516 | 0.033 |
| Product innovation (2005) | 0.000 | 0.389 | 1.784 | 0.310 |
| Product innovation (2006) | 0.000 | 0.375 | 1.625 | 0.313 |
| Product innovation (2007) | 0.000 | 0.369 | 1.956 | 0.317 |

## 4. Empirical Analysis

### 4.1. Verify Selectivity Bias

By using the GPS method, this paper first estimated the conditional distribution of environmental regulation intensity, established the Fractional Logit model, and estimated the parameters. After all of the covariates were selected into the model, the stepwise regression method was used. When the AIC value was the smallest possible value, there were three variables left: labor productivity, capital intensity, and firm size. The model estimation results are presented in Table 3. The significance of the model verifies the existence of selective bias and mixed bias, especially the selective bias, which cannot be solved by the general regression model. According to the estimation results, the labor productivity, capital intensity, and size of an industrial enterprise determine the intensity of environmental regulation of the enterprise to a certain extent. That is, when the government stipulates that the pollutant discharge fee should be levied on the pollutant discharge, the enterprise will make adjustments according to its own state of operation and capital. If the production efficiency is not changed, then the sewage discharge fee will be paid and the production cost will increase. Under these circumstances, whether the enterprise can maintain good operations depends on the characteristics of the enterprise itself. An industrial enterprise with low labor productivity and concentrated capital has a large probability of selecting a large amount of sewage to maintain labor productivity because it has enough capital to pay sewage fees. Conversely, a company with a high labor productivity level

and lack of capital will choose to reduce the amount of pollutants. This is because its own leverage is high, capital is difficult to obtain to support the large scale of the company, and the increase in sewage charges will make the capital turnover more difficult. This is strong proof of the existence of selective bias. To a certain extent, companies "self-select" the intensity of environmental regulation imposed by the government. In this case, it is unreasonable to select the matching variable along with the treatment variable in the right end of the regression model in the previous study, as it would cause endogenous and reverse causal problems.

By using the results for parameter estimation, the propensity score of each individual in the sample can be calculated to prepare for the subsequent implementation of the GPS method.

**Table 3.** Maximum likelihood estimation.

| Matching Variable | Mean | Standard Deviation | Z Statistic | *p* Value |
|:---:|:---:|:---:|:---:|:---:|
| Intercept | −0.158 | 0.581 | −0.272 | 0.785 |
| Productivity | −0.297 | 0.102 | −2.919 | 0.004 ** |
| Intensity | 0.311 | 0.086 | 3.634 | 0.000 *** |
| Size | −0.139 | 0.057 | −2.434 | 0.015 * |

***, ** and * respectively, represent significant coefficients at the 0.1%, 1%, and 5% and 10% levels.

*4.2. Balance Test*

Before performing the last two steps of the GPS method, a balance test of the Fractional Logit model results is required. An individual with the same generalized propensity score value should be considered "equal", so that the generalized propensity score matching method is reasonable in this paper, and the reliability of the results is high. For continuous treatment variables, one method of testing balance is to perform regression on the treatment variable for each covariate before and after matching. The regression after matching needs to add a generalized propensity score to one end of the explanatory variable. The *t*-test value of the regression coefficient can show whether the covariates are varied at different treatment levels. Therefore, the balance can be checked by comparing the *t*-test values before matching and after matching. Figure 1 shows the results of the balance test, where each point represents a covariate, the open point represents the *t*-test case before matching, and the solid point represents the case after matching. It is apparent from Figure 1 that almost all the t-test values after matching are closer to 0 than those before matching. This shows that the degree to which each matching variable determines the treatment variable after matching is greatly reduced, and the "self-selection" problem is well solved, or rather, the balance test is good.

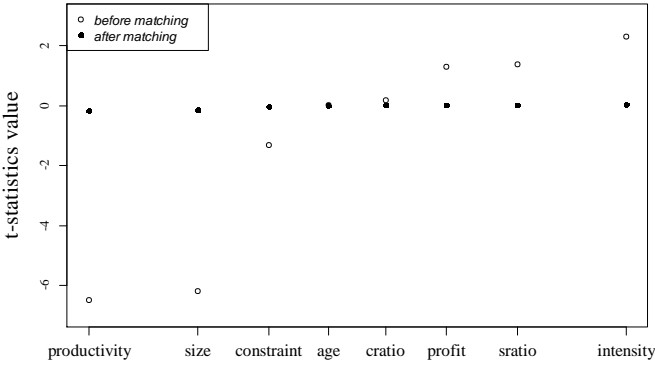

**Figure 1.** The balance test.

*4.3. Estimation of Causal Effect of Environmental Regulation Intensity on Technological Innovation Level of Enterprises*

Based on good balance, the high-order regression equations of environmental regulation intensity and generalized propensity score are used to fit the results variable, process innovation, and product

innovation level. Considering the lag of enterprise technological innovation, this paper, respectively, fits the level of enterprise technological innovation that lags one to three years. The regression results are presented in Table 4. Using the results of stepwise regression, the high-order terms and interaction terms in the regression equations with different lag periods are introduced separately.

**Table 4.** Least squares estimators.

| | Lag = 1 | | Lag = 2 | | Lag = 3 | |
|---|---|---|---|---|---|---|
| | **Process Innovation** | **Product Innovation** | **Process Innovation** | **Product Innovation** | **Process Innovation** | **Product Innovation** |
| Intercept | 0.0151 *** | $<2 \times 10^{-16}$ *** | −0.0135 * | $<2 \times 10^{-16}$ *** | 0.0052 | $<2 \times 10^{-16}$ *** |
| $gps$ | | 0.0002 *** | 0.6567 *** | 0.0104 * | 0.1820 | 0.0021 ** |
| $T$ | −0.1126 *** | 0.3356 | −0.0081 | 0.0031 ** | | 0.0007 *** |
| $gps \times T$ | 0.8109 *** | | | | | |
| $gps^2$ | | 0.0002 *** | −4.2141 *** | 0.0135 * | −0.8361 | 0.0057 ** |
| $T^2$ | 0.0389 | 0.8243 | | 0.0181 * | | |
| $T^3$ | | 0.8692 | | | | 0.0269 * |
| $gps^3$ | 0.9590 *** | 0.0002 *** | 9.6849 *** | 0.0149 * | 2.8320 *** | 0.0139 * |

***, ** and * respectively, represent significant coefficients at the 0.1%, 1%, and 5% and 10% levels.

The variation range of environmental regulation intensity is divided into a plurality of sub-intervals, and the average treatment effect on the treated (ATT) is calculated in each sub-interval. Finally, the ATT values of each interval are connected by lines, and the causal relationship between environmental regulation and two kinds of technological innovation in the whole interval is obtained. The GPS method controls the difference of covariates well and solves the selective bias. At this time, the change in the level of technological innovation of enterprises can be explained as the causal effect of changes in the intensity of environmental regulation.

Figure 2 depicts the causal effect of the intensity of environmental regulations in two kinds of technological innovation lagging from one to three years. It can be seen from the figure that the impacts of environmental regulations on process innovation and product innovation are considerably different.

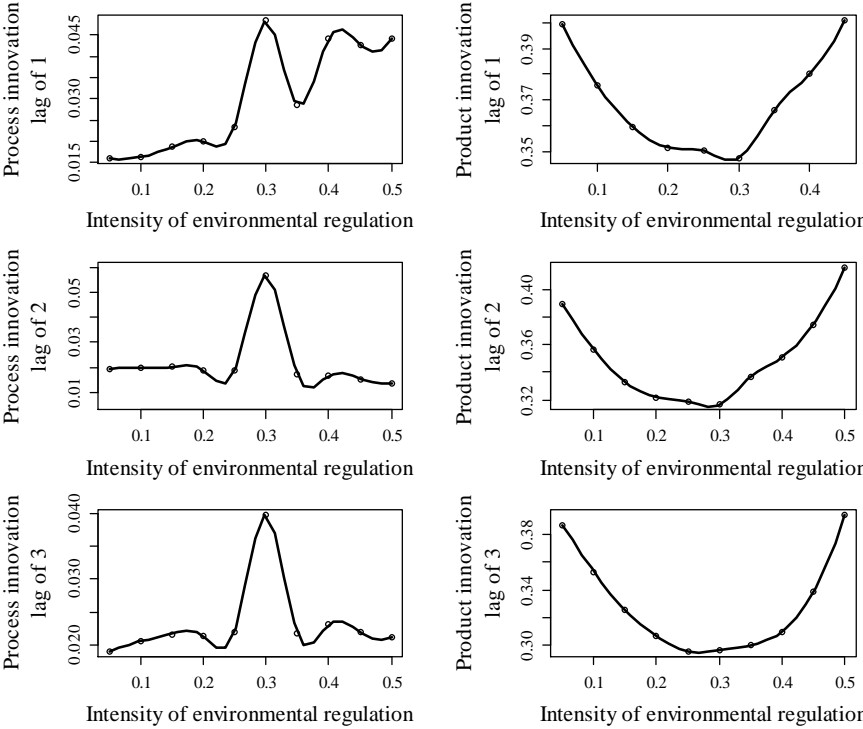

**Figure 2.** The causal relationship between environmental regulation and technology innovation.

The relationship between environmental regulation and the level of enterprise process innovation is not a simple U-shaped or inverted U-type relationship. The relationship between the two is related to the number of lag periods. When the first phase is delayed, the two exhibit an N-type relationship, and there are two inflection points. On the left side of the first inflection point, the increase in the intensity of environmental regulation promotes the process innovation of enterprises. It is worth noting that when the intensity of environmental regulation is small, the role of environmental regulation in promoting process innovation is not obvious, and the relationship between the two is relatively flat. When the intensity of environmental regulation increases to 0.22, the promotion suddenly increases. When the environmental regulation intensity is higher than the first inflection point, the process innovation level decreases as the environmental regulation intensity increases. However, when the environmental regulation intensity continues to increase until the second inflection point (0.35) is broken, Porter's hypothesis begins to be valid again. The situation of the second and third lag phases is different. The intensity of environmental regulation and the level of process innovation are inverted U-shaped, and there is an optimal environmental regulation intensity. The optimal environmental regulation intensity values with a lag of 2 and 3 are all 0.3, but the maximum process innovation level value with a lag of 2 is higher than with a lag of 3. In addition, when the number of lag periods is greater than 1, the environmental regulation intensity can only have a significant impact on the level of process innovation within (0.25, 0.37). Environmental regulations that are too strict or too lenient cannot significantly change the level of process innovation. Therefore, the causal relationship between environmental regulation and process innovation has two dimensions: "period" and "strength". This is consistent with the conclusions of Han et al. [10], but the scholars did not provide a specific dynamic change relationship. This paper considers that, in the time dimension, the relationship between the two changes from N to inverted U, which is a nonlinear relationship. When the first phase is delayed, there are two inflection points. When the lag is more than 1 period, there is only one inflection point, that is, the optimal environmental regulation intensity.

The impact of environmental regulation and enterprise product innovation is relatively straightforward, and it is not related to the number of lag periods. The relationship between the two is U-shaped with a threshold of 0.3, which is the same as the first inflection point value of environmental regulation affecting process innovation. This shows that if we want to achieve the same growth of product innovation and process innovation, it is necessary to set the environmental regulation intensity to be more than 0.35 when the first phase is delayed, but this growth cannot be achieved when the lag is more than 1 period. In summary, the establishment of the Porter's hypothesis is conditional, and the impact of environmental regulation on the level of two kinds of technological innovation is nearly opposite.

### 4.4. Discussion

Combined with the existing research and the actual situation, we have the following appropriate explanations for the conclusions of the above model. When the government implements environmental regulation, it takes a certain amount of time for companies to respond to the policy, which can be seen in the graph as the incentive driven by environmental regulation at almost zero when the intensity of environmental regulation is small. As the intensity of environmental regulation increases, companies realize that passive pollution control is not a long-term strategy. To maintain long-term competitiveness in the market, it is necessary to try to develop innovation within the production process. Of course, any regulatory policy should be moderate. When the intensity of environmental regulation is large enough, although enterprises have enough motivation to carry out process innovation, it takes time. If the speed of process innovation cannot catch up with the intensity of environmental regulation, then the situation on the right side of the inflection point will occur. It is worth noting that the causal relationship with a lag of 1 has a second inflection point of 0.35. This may be because when the intensity of environmental regulation increases to a higher level, the enterprise has formed a stable tolerance to environmental regulation and is able to maintain intensive management and sanitary production.

Compared to process innovation, the impact of environmental regulation on product innovation is completely different. Process innovation and product innovation are two different links in the process of industrial production and management. Process innovation focuses on innovations in production processes and the use of new materials, whereas product innovation is an innovation in the products or services provided. Process innovation occurs before the production of a product or service, and it is the main link that produces environmental pollution, but product innovation occurs at the end of the production chain. Both forms of innovation require research and development funding, so environmental regulation has almost the opposite effect on the two kinds of technological innovation. Based on this, the author believes that the impact of environmental regulation on product innovation may be based on process innovation and, therefore, the government should consider the impact on the two kinds of innovation when setting environmental policies.

### *4.5. Robustness Test*

To test stability, this paper divided industrial enterprises into light industry and heavy industry, and it measured the causal effect of the intensity of environmental regulation on enterprise process innovation. Using the lag of one year as an example, the two inflection points of light industry are 0.15 and 0.2, and the two inflection points of heavy industry are 0.31 and 0.37, respectively. The functional relationship between the two is consistent in shape, an N-shape, thereby indicating the robustness of the conclusion. In addition, it can be seen from Figure 3 that the two inflection values of heavy industry are greater than those of light industry. Before the first threshold, the causal line of light industry has a larger slope, and the effect of environmental regulation to promote process innovation is more obvious. This shows that light industry is more likely to achieve a "win-win" situation.

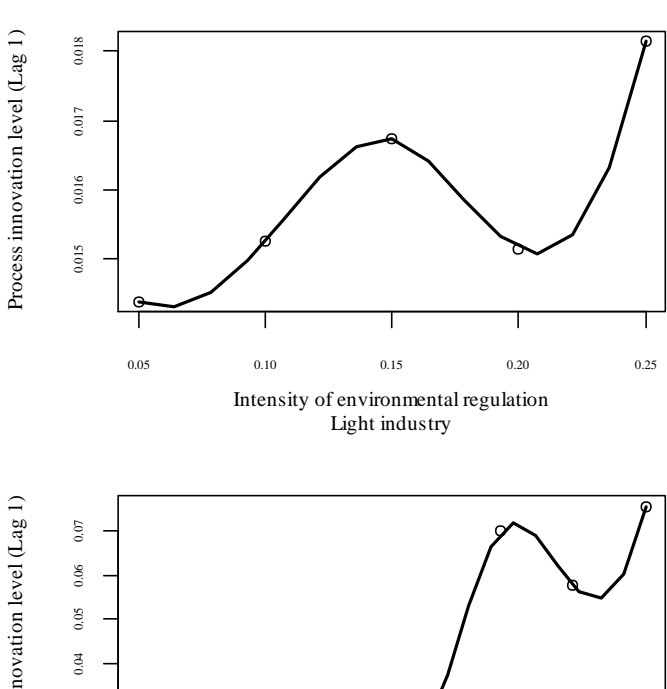

**Figure 3.** The robustness test.

## 5. Conclusions and Implications

In the context of the new era, the Chinese central government emphasizes the need to change the pattern of economic growth and achieve sustainable economic development. This paper focuses on a feasible path of sustainable development—the role of environmental regulation in promoting technological innovation. Based on previous studies, this paper uses the frontier method of a causal inference field—a generalized propensity score matching method—to measure the causal effect of environmental regulation on two kinds of technological innovation. The advantage of this paper is that it solves the selective bias present in in previous studies, weakens the endogenous and reverse causal effects of the model, and obtains more stable and complete conclusions.

### 5.1. Conclusions

The important conclusions obtained in this paper are as follows. The influence of environmental regulation intensity on the level of process innovation has two dimensions: "period" and "strength". In the time dimension, the relationship between the two changes from N to inverted U. In the intensity dimension, the relationship between the two is nonlinear. When the first phase is delayed, there are two inflection points in the relationship between the two. When the number of lag periods is greater than one period, there is an inflection point, that is, the optimal environmental regulation intensity. The impact of environmental regulation on the level of product innovation of enterprises is almost the opposite of its impact on the level of process innovation, presenting a single U-shaped relationship.

### 5.2. Policy Implications

Based on the above conclusions, this paper has the following important implications for China's overall planning of economic and social sustainable development: (1) Use the role of environmental regulation in boosting technological innovation to enhance economic sustainability. Based on the calculation results of this paper, the establishment of this sustainable path requires conditions. The government must clearly define what kind of technological innovation it wants to achieve and the period in which it hopes to achieve innovative results. To focus on stimulating companies to carry out process innovation, it is necessary to increase the intensity of environmental regulation to reach (0.25, 0.3), to ensure the significance of incentives. However, the government should not blindly improve the intensity of environmental regulation, or it could easily produce the opposite effect. The optimal environmental regulation intensity is 0.3, and a higher environmental regulation intensity can be set if the first phase is delayed. However, if we want to focus on stimulating enterprises to reform product innovation, we need to break the threshold of 0.3 and maintain the growth on the right side of the threshold. This involves a trade-off between process innovation and product innovation to a certain extent. Therefore, if we want to balance process innovation with product innovation, the government must carefully consider environmental regulation. (2) Implement differentiated environmental regulation. Different industries have different characteristics of pollutant discharge, so the government should divide the industries according to the emission degree of environmental pollution and implement differentiated environmental policies. In order to achieve the goal of innovation-driven sustainable development, heavy industries with high emissions of pollution need to be more effectively restrained than light industries. (3) Rolling revision of environmental policies is necessary. The key to strengthening the role of environmental regulation in promoting technological innovation lies in the effectiveness of environmental policies. The government should maintain a rolling policy revision and track the effect of the policy in real time to determine where the current environmental regulation's role in boosting technological innovation lies in a non-linear relationship, so as to achieve the goal of maximizing economic development while ensuring the quality of the environment. (4) Improve pollution control subsidies and incentive mechanisms. The non-linear relationship between environmental regulation and process innovation shows that when the intensity of environmental regulation is higher than 0.3, the boosting effect disappears and the level of process

innovation declines. At this time, production technology funds are crowded out by sewage costs. If the government can properly provide investment and subsidies to enterprises with high level of environmental regulation, it will help to promote process innovation. When the intensity of regulation is within (0.25, 0.3), although environmental regulation has a driving effect on process innovation, in order to ease the pain caused by regulation and accelerate the promotion of a supportive role, the government should also increase fiscal expenditure to subsidize enterprises and guide social investment to some extent.

### 5.3. Implications for Future Research

The generalized propensity score matching method used in this paper solves the selectivity bias in previous studies well, and obtains different conclusions from previous studies. Most previous studies only found an inflection point, and did not find the opposite effect of environmental regulation intensity on the two kinds of technological innovation. The dynamic change between environmental regulation and technological innovation was not fully discovered in previous studies. The implication of this paper for future research is when exploring the relationship between environment-related variables and economic-related variables, the existence of selectivity bias must be considered. General statistic regression methods cannot solve the endogenous and reverse causal effects caused by selective bias. Causal inference methods such as generalized propensity score matching can be used to eliminate the endogenous and reverse causal effects to obtain more robust conclusions.

### 5.4. Limitations

The limitations of this article mainly stem from the period of the data. In order to discuss the impact of incentive environmental regulation rather than other types of environmental regulation on technological innovation, this paper uses the sewage charges paid by enterprises to measure the intensity of environmental regulation. The database used in this research is the China Industrial Enterprise Database, the most authoritative micro database in China. However, because the database only disclosed the indicator of sewage charges in 2004, the data period of this article is from 2004 to 2007. Older data may have some impact on conclusions. At present, the disclosure of corporate environmental performance in corporate social responsibility reports is extremely incomplete, so it is extremely necessary to develop other better indicators to measure incentive environmental regulation or to obtain high-quality data in the new period, which will be the improvement direction of the next step of research.

**Author Contributions:** Conceptualization, G.-Y.Z. and R.G.; methodology, G.-Y.Z. and R.G.; software, G.-Y.Z.; validation, G.-Y.Z., R.G. and H.-J.W.; formal analysis, G.-Y.Z.; investigation, H.-J.W.; resources, H.-J.W.; data curation, G.-Y.Z.; writing—original draft preparation, G.-Y.Z.; writing—review and editing, H.-J.W.; visualization, G.-Y.Z. and R.G.; supervision, H.-J.W.; project administration, H.-J.W. All authors have read and agreed to the published version of the manuscript.

**Funding:** This research received no external funding.

**Acknowledgments:** Supported by Program for Ministry of Education in China Project of Humanities and Social Sciences under grant No. 18YJC790162 and 18YJA790056 and Innovation Research in Central University of Finance and Economics.

**Conflicts of Interest:** The authors declare no conflict of interest.

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
