# Peer review of "The Nonlinear Causal Relationship Between Environmental Regulation and Technological Innovation—Evidence Based on the Generalized Propensity Score Matching Method"

_sustainability, doi:10.3390/su12010352_

Round 1

Reviewer 1 Report

In my personal opinion, the paper is a nice piece of work. It is well written and well organized, and the analysis contained is interesting and provides a nice contribution to the existing literature. My main comments concern the database used, which in my view leave some room for improvement.

Comments

1) The database used in the study provides data for the period between 1998 and 2013. It would be essential to have information corresponding to a more current period.

2) At the end of the section "4. Empirical analysis " a discussion of the results obtained should be included.

Reviewer 2 Report

Below are my comments on the paper entitled: “Nonlinear Causal Relationship between Environmental Regulation and Technological Innovation—Evidence Based on the Generalized Propensity Score Matching Method”. Although the paper is interesting, there are some major areas of concern. First of all, I think you have missed sone key and recent studies of stakeholder, Environmental sustainability and Innovation that must be utilised to develop your introduction and literature review section.as it stands,  I think the positioning of the paper is not strong and need to be enrich with the current literature.  I think you should frame your introduction more closely around sustainability

 From 1-4, I started seeing some language issues so a service of professional proofreader might be needed.

Methods: I am not strong on this technique, so you need to double-check your analysis and data source.

Conclusions and policy implications: here I think you need a more detailed and robust examination of the managerial and policy implication of the findings. The current version is just too weak. . I also want to see a more robust suggestions for future research.

Many thanks for this opportunity to read the work and hope the comments help to develop it further. 

Below are potentially useful articles.

Amankwah‐Amoah, J., Danso, A., & Adomako, S. (2019). Entrepreneurial orientation, environmental sustainability and new venture performance: Does stakeholder integration matter?. Business Strategy and the Environment, 28(1), 79-87.

Danso, A., Adomako, S., et al.  (2019). Environmental sustainability orientation, competitive strategy and financial performance. Business Strategy and the Environment.

Adomako, S., et al. (2019). Environmental sustainability orientation and performance of family and nonfamily firms. Business Strategy and the Environment.

Danso, A., Adomako, S., Lartey, T., Amankwah-Amoah, J., & Owusu-Yirenkyi, D. (2019). Stakeholder integration, environmental sustainability orientation and financial performance. Journal of Business Research.

Lartey, T., Yirenkyi, D. O.,  et al.  (2019). Going green, going clean: Lean‐green sustainability strategy and firm growth. Business Strategy and the Environment.

Adomako, S., Amankwah‐Amoah, J., & Danso, A. (2019). The effects of stakeholder integration on firm‐level product innovativeness: insights from small and medium‐sized enterprises in Ghana. R&D Management.

Round 2

Reviewer 1 Report

Considering that this database is the only one that can be used for empirical analysis in this paper, the authors must include, in the conclusions section, new text in reference to the limitations of this study.

Reviewer 2 Report

the issues i noted have been addressed.

Author Response

    Thank you again for your valuable suggestions on this study. We checked the full text again to ensure the accuracy of the article.